# Workplace Factors, Burnout Signs, and Clinical Mental Health Symptoms among Mental Health Workers in Lombardy and Quebec during the First Wave of COVID-19

**DOI:** 10.3390/ijerph19073806

**Published:** 2022-03-23

**Authors:** Filippo Rapisarda, Martine Vallarino, Camille Brousseau-Paradis, Luigi De Benedictis, Marc Corbière, Patrizia Villotti, Elena Cavallini, Catherine Briand, Lionel Cailhol, Alain Lesage

**Affiliations:** 1Research Center of the Institut Universitaire en Santé Mentale de Montréal (CRIUSMM), Montréal, QC H1N 3V2, Canada; corbiere.marc@uqam.ca (M.C.); villotti.patrizia@uqam.ca (P.V.); catherine.briand@uqtr.ca (C.B.); 2Research and Development Team, Sociosfera Onlus Società Cooperativa Sociale, Via Antonio Gramsci 8, 20831 Seregno, Italy; 3Department of Brain and Behavioral Sciences, University of Pavia, 27100 Pavia, Italy; martine.vallarino@unipv.it (M.V.); elena.cavallini@unipv.it (E.C.); 4Institut Universitaire en Santé Mentale de Montréal, CIUSSS de l’Est-de-l’Île-de-Montréal, 7401 Rue Hochelaga, Montréal, QC H1N 3M5, Canada; camille.brousseau-paradis@umontreal.ca (C.B.-P.); ldebenedictis.crfs@ssss.gouv.qc.ca (L.D.B.); lionel.cailhol.med@ssss.gouv.qc.ca (L.C.); alesage.iusmm@ssss.gouv.qc.ca (A.L.); 5Research Chair Mental Health and Work, Institut Universitaire en Santé Mentale de Montréal, Montréal, QC H1N 3V2, Canada; 6Department of Education and Pedagogy, Université du Québec à Montréal, Montréal, QC H3A 0E8, Canada; 7Department of Occupational Therapy, Université du Québec à Trois-Rivières, Trois-Rivières, QC G8Z 4M3, Canada

**Keywords:** COVID-19, work context, mental health service, health workers, burnout, depression, anxiety

## Abstract

Several research contributions have depicted the impact of the pandemic environment on healthcare and social care personnel. Even though the high prevalence of burnout depression and anxiety in healthcare settings before COVID-19 has been well documented in the research, the recent increase in psychological distress and mental health issues in healthcare and mental health workers should be attributed to the effect of the COVID-19 pandemic. The aim of the present study is to develop, evaluate, and compare a model of COVID-19 workplace stressors between two different territories, the Italian region of Lombardy and the Canadian province of Quebec. Within this model, burnout is depicted as the strongest determinant of mental health symptoms for mental health workers. In turn, the main workplace determinants of burnout are the perception of a lack of support from the organization and the fear of contracting COVID-19 at work. Findings also provide insights for designing interventions to promote and protect mental health workers in the context of the pandemic. In conclusion, it is necessary to monitor burnout and carefully analyze elements of organizational culture, in addition to offering clinical and psychological care for those in need.

## 1. Introduction

The COVID-19 pandemic poses complex challenges to the public health and social care systems, including mental health services (MHSs). Since the first wave in 2020, in hospital settings, specific wards or areas were created for infected patients, and systems were put in place to sanitize environments [1,2]. In outpatient services, several activities, especially group rehabilitation, were suspended, and many of the monitoring, psychotherapy, and assessment interviews were conducted remotely, through telephone or video calls, at least in the first phase of the outbreak [1,3,4,5]. In addition, there has been an overall deterioration in the mental health of the population of all ages, which has further burdened the system of care [6,7].

The detrimental impact of the pandemic on healthcare and social care personnel wellbeing has already been documented in the past, as reported in a recent review [8], and several research contributions confirmed and extended findings from previous studies for the COVID-19 outbreak. Two meta-analyses, the first including 13 cross-sectional studies and a total of 33,062 participants in the first stage of the pandemic [9], and the latter including 76 studies [10], provided evidence that a high proportion of healthcare professionals have experienced levels of clinical mental health symptoms (such as anxiety, depression, and insomnia) and also evidence of an overall increase in occupational burnout and compassion fatigue. Another survey conducted on a sample of 631 rural paramedics, police officers, community nurses, and child protection workers in Australia [11] found that 27.1% of the sample indicated moderate or severe levels of anxiety, four times higher than the proportion in the general population; 16.5% recorded moderate or severe levels of depression, 10 times higher than the general population; 56.1% scored in the high range for emotional exhaustion (EE), 52% higher than for similar groups pre-COVID-19, and a results showed a slight increase in (professional) depersonalization.

Even though the high prevalence of depression, anxiety, and burnout in healthcare settings before the COVID-19 has been well documented in the research [12,13,14,15], the increase in psychological distress and mental health issues in healthcare workers, including mental health and social care workers, should be attributed to the effects of the COVID-19 pandemic. Several authors noted the role of work-related factors associated with the pandemic situation. First, the pandemic setting required services to modify procedures and redeploying personnel in a relatively short time; thus, adjustment to changes in tasks and increased workloads were common issues [9,16,17,18,19], especially in services that face chronic understaffing, and the application of COVID-19 sanitation and infection control procedures (such as PPE wearing and PPE disposal) were reported as pandemic-related difficulties [18,20,21]. Second, fear of COVID-19 infection significantly impacted occupational mental health [22,23,24] and could be associated with several work-related determinants; being a frontline worker and having direct contact with patients with COVID-19 increased distress and emotional exhaustion [5,11,16,18,25], and lack of personal protective equipment (PPE) was a common concern, especially in the first stages of the pandemic [20,26].

Organizational support and climate also played a role in mediating the effects of the COVID-19 pandemic. The perceived lack of organizational support [11,17,26], dissatisfaction with the quality of communication [11], and conflicts or lack of social contact with colleagues were also associated with increased emotional exhaustion and depression symptoms [11].

Additionally, mental health care settings may present specific challenges and stressors when compared to general healthcare services. Maintaining the therapeutic relationship with people that may be distressed, suspicious, or struggling to comprehend the situation could be complicated with respect to infection control requirements and providing safe care [20]. Staff in inpatient and residential settings, where continuing face-to-face contacts are more frequent than in outpatient care, are more prone to psychological distress [20]. However, other studies suggest that the effect of setting (inpatient vs. outpatient) in mental health care may be more complex [5,27]. Moral distress related to the perception of a decrease in the quality of care was a commonly reported issue in surveys of mental health care staff [18,28].

However, despite the aforementioned evidence, there has not yet been an attempt to develop a comprehensive conceptual model based on data rather than on a testing regression model using single construct, such as anxiety, depression, and burnout dimensions as single dependent variables.

Therefore, the aim of the present study is to develop and evaluate a model of COVID-19 workplace factors (Figure 1) in two different territories and contexts, i.e., the Italian region of Lombardy and the Canadian province of Quebec. The comparison of both environments could be an opportunity to test the model’s validity and to increase its generalizability, while at the same time focusing on specific differences.

We hypothesize that the relationship between workplace factors and common mental health symptoms (anxiety and depression) may be indirect and mediated by fear of the COVID-19 contagion and burnout signs. Moreover, we considered burnout as a determinant of mental health symptoms as reported in the existing literature [29] and in recent studies on healthcare workers during COVID-19 pandemic [30]. Therefore, the impact of work-related factors on fear of contagion and burnout will be explored. For some factors, we may have a previous hypothesis: i.e., lack of PPE could be a determinant of fear of the contagion, but not of burnout; conversely, lack of organizational support could be linked with burnout but not with fear of contagion. However, for most of the factors, the pattern of influence could not be determined a priori, and it could differ in the two environments (Lombardy and Quebec). Therefore, we aim to test this model separately in the two environments, using a path analysis technique. It will then be possible to compare the retained model stemming from each context to the other and establish the degree of generalization. 

## 2. Materials and Methods

### 2.1. Design and Data Collection

The present study is a part of a cross-sectional multicenter survey of workers and professionals working with people experiencing mental health problems in Lombardy (Italy) and Quebec (Canada). Preliminary descriptive findings related to Lombardy have been previously published [5].

Participants were recruited from different outpatient and inpatient services, including community mental health centers, residential facilities, hospital wards, and nursing homes. Upon granting informed consent, the subjects were asked to fill in an anonymous online questionnaire that lasted about 10 min.

In Lombardy, data were collected between 15 April and 15 May 2020. The first COVID-19 cases in Lombardy were officially reported in the middle of February 2020, and the rapid escalation of the outbreak required the Italian government to introduce a mandatory lockdown on 7 March 2020. By May 15, 84,119 cases of COVID-19 infection (circa 8411 cases per one million inhabitants) and 14.411 COVID-19 related deaths (circa 1450 per one million inhabitants) were reported. An online platform was used to collect data, and recruitment was carried out by spreading a link to an invitation to participate through various professional networks. A “snowball” dissemination strategy was promoted, in which each participant was asked to send the survey invitation link to their colleagues. Details of the Italian data collection can be found in a previous publication [5].

In Montreal, data were collected during the second half of Quebec’s first COVID-19 wave, from July to October 2020. According to the Quebec public health institute, the first COVID-19 wave ended on 11 July 2020, but public health measures remained in place for distancing, mask wearing in public places, limits to private gatherings, and distance working for public employees; universities also announced the continuation of virtual courses for autumn. The second wave started 23 August 2020, but public health measures were not increased significantly until November. By 8 June 2020, Quebec already reported 5000 deaths (circa 625 per 1 million inhabitants). The number of daily new cases increased from 12 to 950 (1.5 to 119 per 1 million inhabitants) from the beginning of July until the beginning of October, the ending point for our data collection. Recruitment was mainly carried out by the Mental Health and Addiction Program (PSMD) of the CIUSSS-EMTL. The management of the program sent the survey invitation link by email to the 400 employees and physicians (using an institutional email address) working with mental health patients. The head of the CIUSSS-EMTL psychiatry department sent the invitation by email to all the psychiatrists of the department. With the help of the Ordre des infirmiers et infirmières du Québec (OIIQ), the PSMD management also sent the invitation to the 158 nurses of the CIUSSS-EMTL who had consented to be solicited for research projects.

### 2.2. Instruments

The survey consisted of two sections—developed to profile participants’ occupational and job characteristics—to identify COVID-19-related work factors that might affect psychological well-being, and to assess burnout and common mental health problems. All items were formulated in parallel in Italian and French by the research team composed of native Italian and French-Canadian speakers with proficiency in both languages.

In the first section, ad-hoc items collected socio-demographic, professional background, and working conditions during the pandemic. Moreover, self-reported narratives of work-related difficulties were collected through an optional open-ended question (“What are the main difficulties you have encountered during the last four weeks in your professional activity?”) to determine the causes of the discomfort affecting the participants in their job activity. The participants could freely answer the question, in their own words, in an assigned space without word or time limits. The research team, composed of Canadian and Italian researchers, created a first set of thematic codes after reading all open-ended questions in the Canadian and Italian samples. The coding system was designed to code each single answer using one or multiple thematic codes and was pilot tested in both samples (20 answers per database); some additional modifications to the coding set were implemented. The final coding set was composed of 32 thematic codes grouped in seven areas, i.e., Organizational Support, Job Design, Emotional Distress, Perception of Decreased Quality of Care, Teamwork and Group Climate, Work-Life Balance, and Restrictions of Safety Rules. Final codes were computed by selecting, for each case, only thematic codes that were coded by the two coders (agreement). The participants’ answers to the open-ended question were extracted from an independent database to facilitate analysis. One of the researchers (M.V.) performed a content analysis to identify the emerging categories for which all answers could be coded. Categories and reliability indexes (Cohen’s Kappa) are presented in the Appendix A.

The second section included three validated questionnaires on psychological distress (the Maslach Burnout Inventory, Generalized Anxiety Disorder–7 and Patient Health Questionnaire-9). 

The Maslach Burnout Inventory (MBI) [31,32,33,34] is a self-administered instrument to assess burnout in health organizations through three subscales, i.e., Emotional Exhaustion (EE), Depersonalization (DP), and (reduced) Personal Accomplishment. Items are framed as statements of job-related feelings, which are rated on a 7-point frequency scale ranging from 0 (never) to 6 (every day). In the present study, only the Exhaustion and Depersonalization scales were adopted, since the developers of the MBI have expressed some doubts about whether Personal Accomplishment is a valid burnout dimension [35,36].

The Generalized Anxiety Disorder-7 questionnaire (GAD-7) [37,38] contains seven items, and it has been the most frequent used instrument to assess anxiety levels in the general population [39,40] and also among healthcare workers during the COVID-19 pandemic [9]. 

The Patient Health Questionnaire-9 (PHQ-9) [41] is a validated questionnaire that assesses the presence of depressive symptoms among patients [42], and it has been frequently adopted to assess psychological distress in the general population [39] and among healthcare workers during the COVID-19 pandemic [9]. 

### 2.3. Data Analysis

Descriptive statistics were computed for sociodemographic and work-related variables, and the differences between the Lombardy and Quebec settings were assessed through parametric and nonparametric tests according to the nature and distribution of the variables analyzed.

Some response categories were merged during data analysis to facilitate data analysis and results presentation. Concerning the “setting” variable, the label “outpatient service” was adopted for community mental health centers, office-based activities, counselling and psychotherapy services, day hospitals, and day-care units, whereas the label “inpatient service” included hospital wards, residential facilities (including those from high-intensity services and those from low-intensity services, without staff onsite), and nursing homes. Moreover, for the professional role variable, the label “counselor” was adopted for rehabilitation workers with a university degree, i.e., professional educators, occupational therapists, and rehabilitation technicians. 

To obtain comparable estimates of burnout and distress symptoms in the two samples, we estimated the number of participants that showed severe levels in MBI’s EE scale (EE > 22 for the Italian version [33]; EE > 28 for the French-Canadian version [34]) and DP scale (EE > 6 for the Italian version [33]; EE > 8 for the French-Canadian version [34]) and moderate (GAD-7 ≥ 10) and severe (GAD-7 ≥ 15) levels of anxiety [37] and depression (moderate, PHQ-9 ≥ 10; severe PHQ-9 ≥ 15) [42]. 

The conceptual model of work-related determinants of distress, fear of COVID-19 infection, burnout signs, and mental health symptoms were tested by running a path-analysis using the R package lavaan to develop models to explore the relationship between common mental health symptoms (measured with GAD7 and PHQ9), burnout signs, and work-related variables. The analysis proceeded through two steps. First, a complete model including all the variables was tested. Three latent factors were included: Fear of Contagion (FoC), composed by ad hoc items “perceived probability of contracting COVID-19 at work,” “concern for contracting COVID-19 at work,” and “concern for infecting users at work;” Burnout Signs (BS) composed of the Emotional Exhaustion and Depersonalization MBI scales; Common Mental Health Symptoms, composed of the GAD7 and PHQ9 scores. The following COVID-19-related workplace factors were included: Setting (inpatient vs. outpatient), Availably of Personal Protection Equipment (PPE), Increased Workload, and clusters total scores from the Self-Perceived Work-Related Difficulties section, i.e., Organizational Support, Job Design, Emotional Distress, Perception of Decreased Quality of Care, Teamwork and Group Climate, Work-Life Balance, and Restrictions of Safety Rules. In the next step, final model tuning was achieved by eliminating variables and regression paths that were non-significant (*p* > 0.05), and additional paths were added considering modification indexes. Models were tested separately in the two datasets (Lombardy and Quebec) due to the presence of major differences in work related context, time, and data collection procedures. For each model, the assessment of the fit was based on the chi-square test, the Comparative Fit Index (CFI; with the criterion of 0.95), the Tucker–Lewis Index (TLI; with the criterion of 0.95) [43], and the root mean square error of approximation (RMSEA; with the criterion of 0.05 as upper limit to consider a good fit) [44].

All data analysis was performed using IBM SPSS Statistics v. 23 and R.

## 3. Results

### 3.1. Participants’ Characteristics

Of the 608 (337 in Lombardy and 271 in Quebec) participants who began the online survey, 221 dropped out, and 121 in Lombardy (35.9%) and 87 in Quebec (32.1%) were excluded from the analysis. Therefore, analyses were performed on a sample of 396 participants, 212 in Lombardy and 184 in Quebec. 

Sample descriptors are shown in Table 1. The most represented professional categories were psychologist and rehabilitation counselors in Lombardy, whereas, in Quebec, they were social workers, nurses, and rehabilitation counselors. In both samples, most of the participants worked in outpatient services and were female, with a slight difference in the mean age (about 44 vs. 42). 

Regarding the working conditions, most of the participants reported an adequate availability of PPE, but, compared to the Lombardy sample, the Quebec group reported a significantly lower level of availability.

In both samples, burnout signs were more common than symptoms of mental health deterioration (Table 2). Moderate to severe emotional exhaustion and depersonalization were detected in more than half of the surveyed workers in both settings, with higher levels in Quebec, especially regarding exhaustion. Apart from very few cases (0.5–1.4%), clinical levels of mental health symptoms were always present in combination with burnout signs.

The most common difficulties reported by MHWs in both samples are related to Job Design, Organizational Support, and Deterioration in the Quality of Care. Significant differences between the two samples were also detected: in Quebec, Job Design issues due to increased workload, lack of Organizational Support related to lack of moral support and an authoritarian management style, and physical discomfort toward mask and safety procedures were reported more frequently as compared to the Lombardy sample. For the Lombardy sample, issues related to Deterioration of the Quality of Care were more frequent when compared to the Quebec sample.

### 3.2. Path Analysis

Chi-square tests and fit indexes for the initial path analysis models indicated a mediocre fit (Lombardy: chi-square = 120.23; df = 81; *p* <0.01; CFI = 0.93; TLI = 0.91; RMSEA = 0.05; Quebec: Chi-square = 12.87; df = 81; *p* <0.01; CFI = 0.92; TLI = 0.90; RMSEA = 0.05). However, several coefficients’ paths were non-significant, and modification indexes suggested testing for new associations. This was done not only to gain a purely statistical advantage and improve the fit of the model tested, but also because these relations were considered relevant from a theoretical explanatory point of view. 

After removing non-significant associations and adding suggested paths, the fit indexes in both models improved (Lombardy model: chi-square = 53.38 df = 41, *p* = 0.09; CFI = 0.98; TLI = 0.97; RMSEA = 0.04; Quebec model: chi-square = 52.49 df = 42, *p* = 0.13; CFI = 0.98; TLI = 0.97; RMSEA = 0.04). The final path analysis model with standardized regression coefficients is shown in Figure 2 and Figure 3 and in Table 3.

In both models (Lombardy and Quebec samples) the three hypothesized latent factors, i.e., Fear of COVID-19 Contagion (FCC), Burnout Signs (BS), and Common Mental Health Symptoms (CMHS), were confirmed. Burnout signs had the strongest association with common mental health symptoms (β = 0.65, *p* < 0.01 in Lombardy; β = 0.87, *p* < 0.01 in Quebec), Fear of COVID-19 had a moderate but significant association with BS in both samples (β = 0.20, *p* < 0.01 in Lombardy; β = 0.30, *p* < 0.01 in Quebec) and a moderate association with CMHS in Lombardy only (β = 0.23, *p* < 0.01). FCC was mildly associated with Unavailability of PPE (β = 0.33, *p* < 0.01 in Lombardy; β = 0.16, *p* < 0.01 in Quebec) and Lack of Organizational Support was positively associated, in both samples, with BS. Working in an Inpatient Service was associated with FCC in Quebec and with BS in Lombardy. Restriction and Safety Issues had a slight but significant association with Concerns of Contracting COVID-19 at Work (β = 0.16, *p* < 0.05). In the Lombardy model, BS had a slight but significant association with Concern of Transmitting COVID-19.

## 4. Discussion

The current study attempted to develop a conceptual model of the effects of the COVID-19 pandemic-related workplace factors using a path analysis technique. Workplace factors have a direct impact on individual fear of contagion and on burnout signs (emotional exhaustion and depersonalization); in turn, fear of contagion and burnout signs have a detrimental effect on MHWs mental health, resulting in clinical symptoms. Moreover, despite some minor differences, the resulting models were similar in two different contexts, increasing the validity and generalizability of the results and providing useful insights into the field of occupational mental health prevention in the context of services that provide mental health interventions. 

First, findings confirm the role of burnout as a determinant of mental health symptoms for mental health workers, as previously reported in the literature [15,45,46,47]. For MHWs that participated to the present study in Lombardy and Quebec in the first stage of the COVID-19 pandemic, work-related factors may directly increase burnout, and that was also the case for lack of organizational control, as well as, indirectly, for increasing the fear of contracting at work. This was the case for lack of PPE and having colleagues that had already caught COVID-19; setting, i.e., working in an inpatient service, played a significant role in both samples, but with different paths. In Quebec, it directly affected burnout, whereas in Lombardy, its effect was mediated by the fear of COVID-19.

Our findings partially confirm the relevant detrimental role of the fear of COVID-19 transmission on workers’ well-being, with different paths. We find a direct effect on increasing mental health symptoms, but also a more complex path, in which the fear of contagion increases burnout. This makes us speculate that, on a psychological level, the fear of contagion can have an impact on individual well-being with different mechanisms: (1) it increases the perception of vulnerability and uncertainty, reducing the perception of individual control at work, a well-known factor that increases burnout [47]; (2) it creates a psychophysiological activation of threat that may trigger anxious and depressive mechanisms [48]. Moreover, it must be noted that, since fear of COVID 19 has a stronger association with burnout than with Gad7 and PHQ9 scores, it could be considered a source of work-related psychological strain and adjustment rather than a mechanism of pathological anxiety. Similar findings were found as related to previous H1N1 and SARS pandemics [8] when, the initial phase of the spread of the disease, healthcare staff experienced a loss of the sense of job-related control that increased fatigue, frustration, vulnerability and perceived threat. Thus, the role of fear of COVID-19 within the work context should be theoretically and empirically examined within other individual-level factors, i.e., external locus of control, poor self-esteem, maladaptive coping styles, and emotional intelligence that have been associated with burnout [49]. 

Third, the present study only partially found the role of increased workload and job task change to be a factor in increasing psychological distress. In fact, although the participants’ narratives addressed change in tasks and procedures as a factor of difficulty at work, no statistically significant association with either burnout signs or mental health symptoms were found in the model. It can be hypothesized that the increase of workload and change in work practices, consequent to the MHS reorganization due to pandemic emergency, didn’t affect the burnout and mental health of employees per se, but rather the combination of these changes and the perceived lack of support from the organization and its managers caused these effects. However, in the present study, elements related to job design were retrieved using only self-reported narratives, which may have underestimated the impact of job-related factors such workload and caseload, clarity of role, degree of control, shift work, and time pressures [50]. Similarly, the perception of worsening of care, an element of difficulty that emerged most often in the Lombardy sample, was also a relevant element of the subjective experience of the participants, but without having a particular role in the worsening of psychological distress.

Although the patterns obtained in the two contexts are very similar, slight differences are nevertheless present, and these could be attributable to timing and psychological climate. An overall comparison of the two models suggests that, in Lombardy, FCC is more relevant than in Quebec, where, on the contrary, the theme of burnout seems to be more relevant. First, in Quebec, FCC doesn’t directly affect CMHS. Moreover, in Lombardy, the regression coefficients that explain FCC are higher than in Quebec. In Lombardy, setting has a direct influence on FCC, while in Quebec, it influences BS. We explain these differences by hypothesizing that, in Lombardy, data were collected in an earlier phase compared to Quebec, when the psychological impact of the concern for the contagion was more relevant. Conversely, in Quebec, most of the data were collected in summer at the end of the first wave, when the concern for the spread of the virus was lower (compared to the previous months) and the prolonged adjustment to the workplace factor led to an increased perception of exhaustion and fatigue, making burnout the most relevant psychological aspect. The present study has some limitations and strengths. Many of the variables included in the model were obtained from coding open-ended questions; therefore, they do not have the same degree of validity and reliability as a psychometrically validated instrument. The choice to use this methodology was determined, in the design phase of the research, by the fact that workers in Lombardy and Quebec had never been confronted with the experience of a pandemic; therefore, using a quantitative instrument such as a questionnaire would have forced participants to use a predetermined set of items not necessarily relevant to the pandemic context. On the other hand, the themes that emerged from the open-ended question are consistent with what was found using a similar methodology and in the same time frame in other countries [18,20,51], an element that would provide a point in favor of the validity of the instrument in identifying relevant constructs. Another limitation is the cross-sectional design that doesn’t allow for the evaluation of the impact of the workplace factors across time. Interestingly, the healthcare workers individual distress response is not always the same. For example, Dufur et al. [52] conducted a longitudinal data collection that identified four different trajectories of psychological distress among healthcare workers, i.e., recovered, resilient, sub-chronic, and delayed. Thus, psychological distress responses across time may be more complex when observed through a single snapshot. Finally, focusing on difficulties may have underestimate the role of positive job-related experiences and support resources that may have played a relevant role in fostering workers well-being, as stated theoretically by the Job Demands–Resources Model [50]. For example, Magliano et al. [53] reported that mental health workers, after one year of the pandemic emergency, valued teamwork and social climate among colleagues, reviewed the therapeutic plans, developed new skills related to telepsychiatry, and discovered new personal resources in themselves and in users. The evidence that models emerging from Lombardy and Quebec, developed separately, yielded very similar models and factors could be considered the main strength of the study, increasing the external validity. Previous work by co-authors in the very same settings with regards to professional culture of mental health workers [54], with bi-lingual (Italian and French/English), having lived and worked in Quebec and Italy, added to our ability to capture and discuss the differences. The latter strengthens the internal validity of the qualitative analysis and interpretation.

## 5. Conclusions

In conclusion, the present study investigated the role of burnout as a major risk factor for mental health deterioration among mental health workers in the context of the COVID-19 pandemic. In addition, the role of adaptation to changes in work design in terms of workload and tasks due to the COVID-19 pandemic can be scaled back, shifting the focus to the role that organizational support can play in worker well-being during such a complex historical phase. Attention should also be paid to facilitating greater access to treatment by employees that would benefit from evidence-based interventions for the early detection of burnout symptoms and clinical levels of anxiety/depression. Therefore, we believe the present study suggests that when designing interventions to promote and protect mental health workers in the context of the pandemic, particularly with regard to symptoms of anxiety and depression, it is necessary to monitor and reduce burnout and to carefully analyze elements of organizational culture. 

## Figures and Tables

**Figure 1 ijerph-19-03806-f001:**
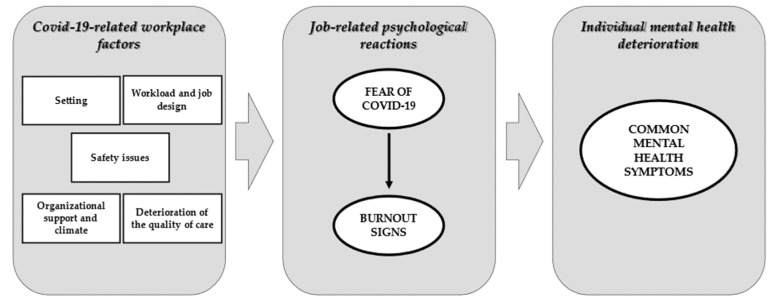
Hypothesized conceptual model of COVID-19 stressors, burnout signs, and clinical mental health symptoms.

**Figure 2 ijerph-19-03806-f002:**
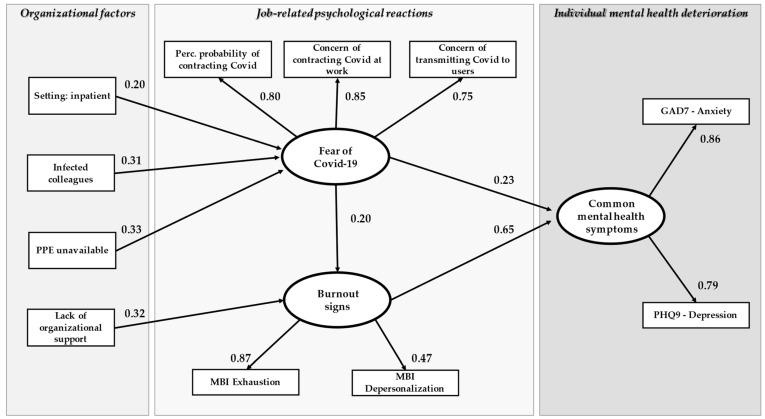
The path analysis model, Lombardy sample. Rectangular boxes = observed variables; oval boxes = latent variables; arrows = regression paths; number = standardized parameter estimates.

**Figure 3 ijerph-19-03806-f003:**
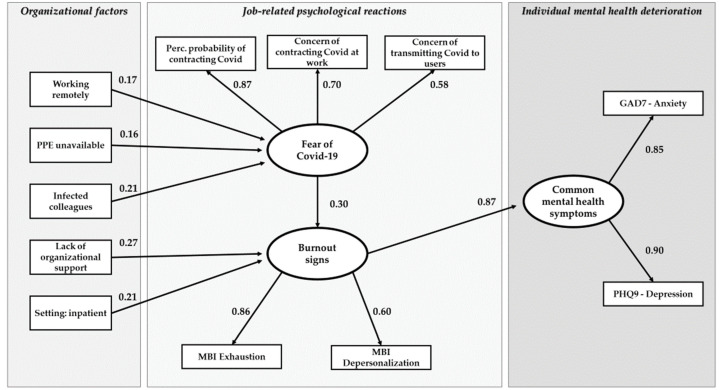
The path analysis model, Quebec sample. Rectangular boxes = observed variables; oval boxes = latent variables; arrows = regression paths; number = standardized parameter estimates.

**Table 1 ijerph-19-03806-t001:** The sample description, work related variables, and psychological distress.

Variables	Lombardy *n* = 212	Quebec *n* = 184
**Sex**		
Female	161 (75.9%)	138 (75.0%)
**Age**		
Mean, sd *	44.4 (12.2)	41.8 (11.1)
**Professional Role** **		
Rehabilitation counselor	62 (29.2%)	39 (21.2%)
Psychologist	59 (27.8%)	14 (7.6%)
Nurse	22 (10.4%)	42 (22.8%)
Medical Doctor	22 (10.4%)	15 (8.2%)
Social Worker	14 (6.6%)	43 (23.4%)
Peer Supporter	15 (7.1%)	0 (0.0%)
Manger or Administration Officer	7 (3.3%)	17 (9.2%)
Other	11 (5.2%)	13 (7.1%)
**Setting**		
Outpatient	122 (57.5%)	110 (59.8%)
Inpatient	73 (34.4%)	74 (40.2%)
**Availability of PPE** *		
Always available or not necessary	192 (90.6%)	145 (78.8%)
Sometimes unavailable	20 (9.4%)	39 (21.2%)
**Did any of your colleagues get COVID-19?**		
Yes	117 (55.2%)	119 (64.7%)
**Did any of your users/clients get COVID-19**		
Yes	107 (50.5%)	109 (50.2%)
**Have you been working remotely only?**		
Yes **	68 (32.1%)	22 (12.0%)
**Did workload increase during COVID-19 pandemic?**		
Yes **	43 (20.5%)	85 (46.2%)

* = *p* < 0.05; ** = *p* < 0.01.; PPE = Personal Protection Equipment.

**Table 2 ijerph-19-03806-t002:** Psychological distress and work-related difficulties.

Variables	Lombardy *n* = 212	Quebec *n* = 184
** ** **Fear of contagion**		
Perceived risk of contracting COVID-19 at work ^a^	1.6 (0.7)	1.5 (0.6)
Concern of contracting COVID-19 at work ^b^ *	1.1 (0.9)	1.3 (0.7)
** ** **Maslach Burnout Inventory**		
Staff above the “moderate emotional exhaustion” cut off	45 (21.2%)	52 (28.3%)
Staff above the “severe emotional exhaustion” cut off	36 (17.0%)	66 (33.0%)
Staff above the “moderate depersonalization” cut off	41 (21.1%)	49 (26.6%)
Staff above the “severe depersonalization” cut off	46 (23.7%)	52 (28.3%)
** ** **Common mental health symptoms**		
GAD7 (anxiety symptoms), mean score (SD)	5.1 (3.4)	4.9 (4.4)
Staff above the “moderate anxiety” cut off	28 (13.2%)	31 (16.9%)
Staff above the “severe anxiety” cut off	3 (1.4%)	7 (3.8%)
PHQ9 (depression symptoms), mean score (SD)	4.8 (2.9)	5.5 (4.5)
Staff above the “moderate depression” cut off	15 (7.0%)	32 (15.3%)
Staff above the “severe depression” cut off	2 (0.9%)	8 (2.3%)
** ** **Self-reported narratives of work-related difficulties**		
Changes in workload and tasks **	61 (30.4%)	78 (42.6%)
Perceived lack of organizational support **	41 (20.1%)	68 (37.1%)
Pandemic context reduced the quality of mental health care **	87 (42.9%)	39 (21.3%)
Emotional distress	27 (13.4%)	35 (18.8%)
Difficulties in respecting safety norms *	12 (5.8%)	22 (11.9%)
Lack of support from colleagues	14 (6.8%)	14 (7.4%)
Work-life balance	14 (6.8%)	5 (2.5%)

* = *p* < 0.05; ** = *p* < 0.01.; ^a^ = item ranged from 1 (low probability) to 3 (high probability); ^b^ = item ranged from 0 (no concern) to 3 (high concern); GAD7 = General Anxiety Disorder 7 questionnaire; PHQ9 = Patient Health Questionnaire 9.

**Table 3 ijerph-19-03806-t003:** The standardized estimates of regression parameters for the Lombardy and Quebec models.

	Lombardy				Quebec			
Regression Parameter	Std. Estimate	Std. Err	z	*p*	Std. Estimate	Std. Err	z	*p*
Determinants of FCC								
LPPE → FCC	0.33	0.06	5.28	<0.01	0.16	0.08	2.08	0.03
Inf Coll → FCC	0.31	0.06	4.83	<0.01	0.22	0.08	2.75	<0.01
Setting → FCC	0.20	0.07	2.85	<0.01	Not included in the model			
Work rem → FCC	Not included in the model				0.17	0.08	2.14	0.03
Determinants of BS								
FCC → BS	0.19	0.08	2.37	0.02	0.30	0.08	3.68	<0.01
LOS → BS	0.32	0.07	4.50	<0.01	0.27	0.07	3.67	<0.01
Setting (inpatient) → BS	Not included in the model				0.21	0.07	2.87	<0.01
Determinants of CMHS								
FCC →CMHS	0.24	0.07	3.36	<0.01	Not included in the model			
BS →CMHS	0.64	0.08	8.45	<0.01	0.87	0.04	21.20	<0.01

FCC = Fear of COVID-19 contagion; BS = Burnout Signs; CMHS = common mental health symptoms.

## Data Availability

The data presented in this study are available on request from the corresponding author.

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
