# Peer review of "Workplace Factors, Burnout Signs, and Clinical Mental Health Symptoms among Mental Health Workers in Lombardy and Quebec during the First Wave of COVID-19"

_ijerph, 2022, doi:10.3390/ijerph19073806_

Round 1
Reviewer 1 Report
The whole research design was clear and the quantitative analysis was well-performed.
The sample size is small for each, how about merge them together as a unified dataset? What is the rationale to describe the results seperately based on regions? Is the sample size big enough to support a complex path model?
If there were strong reasons that these two datasets should be seperated, then the discussion part should further address their differences and similarities based on the contexts.
The result part can be further improved by adding some regression tables, to make the evidence more sufficient.
The limitation and strength part only included some limitations, you can add more statements on its strengths.
Author Response
First, my colleagues and I would like to express our gratitude to the reviewers for taking the time to evaluate our manuscript and for the suggestions that we believe fostered an improvement in the quality of the paper.
Here are authors’ responses to reviewer’s remarks. In addition, we detected some errors in the references numeration, that we corrected.
Comment r1/01. The sample size is small for each, how about merge them together as a unified dataset? What is the rationale to describe the results seperately based on regions? Is the sample size big enough to support a complex path model?
If there were strong reasons that these two datasets should be seperated, then the discussion part should further address their differences and similarities based on the contexts.
Answer. We weighed the pros and cons of keeping the samples separate or combining them, and ultimately felt that the best solution was to analyze them separately for different reasons. First, there are differences in mental health systems and pandemic kinetics. For instance, a fundamental part of the mental health care system in Italy relies on private non profit organizations. In addition, testing the same model on two contexts is one element in assessing its generalizability. In terms of numerosity, the reference is that of Kline (Kline, 1998) who suggests 10 cases per construct. Therefore, for a model like ours, a sample of 150 subjects per model should be sufficient.
We also added a small paragraph about differences between the two emerged models in the discussion.
Comment r1/02.The result part can be further improved by adding some regression tables, to make the evidence more sufficient.
Answer. We included regression tables obtained with the path analysis command in the two samples.
Comment r1/03.The limitation and strength part only included some limitations, you can add more statements on its strengths.
Answer. Some sentences about strengths were added at the end of the discussion paragraph.
Reviewer 2 Report
Introduction: Has understaffing contributed to increased burden and consequent mental health deterioration among health care workers? What about stress associated with rigorous sanitation / PPE wearing? Also, additional risk of aggression / un-compliance with protocols, particularly in emergency settings?
In the "we hypothesize" paragraph - a little more explanation of the two models being tested is required. And, is "workplace stressors" different to "workplace factors"?
Methods: A bit more context around the Covid numbers in Lombardy and Quebec during data collection at each site would be helpful.
In the "to obtain comparable estimates of burnout and distress symptoms" paragraph why were EE cutoffs different for Italian and Canadian samples?
Was some kind of Covid numbers variable included? It seems like that would be a huge predictor of fear of contagion...
Results: Figure 1 is not referenced in the text.
There is no figure 3 (despite reference in the text)
A bit more clarity on the two variations of model tested would be welcome in the results section.
Author Response
Comment r2/01.Introduction: Has understaffing contributed to increased burden and consequent mental health deterioration among health care workers? What about stress associated with rigorous sanitation / PPE wearing? Also, additional risk of aggression / un-compliance with protocols, particularly in emergency settings?
Answer. Unfortunately, we didn’t find relevant literature references with data that link understaffing and aggression issues to staff distress during the Covid19 pandemic. Nevertheless, in the literature that we use for the introduction, there are qualitative findings related to PPE wearing and sanitation and we reported them in the introduction. In addition, they also emerge in the Self-reported narratives of work-related difficulties.
Comment r2/02.In the "we hypothesize" paragraph - a little more explanation of the two models being tested is required. And, is "workplace stressors" different to "workplace factors"?
Answer. We included a couple of sentences in the "we hypothesize" paragraph” that clarifies the scope of the modelling: we didn’t have strong hypothesis for the pattern of workplace factors, so our aim was to develop models to explore their role. We replace “workplace stressors” with “workplace factors” since we prefer a more neutral term.
Comment r2/03.Methods: A bit more context around the Covid numbers in Lombardy and Quebec during data collection at each site would be helpful.
Answer. We added official data about Covid-19 contagion and deaths in the “Design and data collection” paragraph.
Comment r2/04. In the "to obtain comparable estimates of burnout and distress symptoms" paragraph why were EE cutoffs different for Italian and Canadian samples?
Answer. We used the cut off provided by the validation studies of Maslach Burnout Inventory in the two countries (Italy – Sirigatti and Stefanile, 1993; Quebec – Dion and Tessier 1994). Different cutoffs could be considered as a result of cultural, linguistic and contextual factors.
Comment r2/04. Was some kind of Covid numbers variable included? It seems like that would be a huge predictor of fear of contagion...
Answer. We didn’t include Covid numbers in the analysis because the study’s aim was testing workplace factors and not broader contextual factors like covid numbers. Moreover, we should have adopted a multilevel approach to modelling that may didn’t fit in the path analysis. However, we added Covid numbers into the methods section, and we take them into account in the discussion as a possible explanatory factors of the differences between the two models.
Comment r2/05. Results: Figure 1 is not referenced in the text. There is no figure 3 (despite reference in the text).
We added Figure 1 reference in line 90. We made a mistake in titling the figures in the caption. The number has been corrected in the text, therefore:
- figure 1: theoretical model in the introduction;
- figure 2: results, Lombardy;
- figure 3: results, Quebec
Comment r2/06.A bit more clarity on the two variations of model tested would be welcome in the results section.
Answer. We included a regression table in the results section to emphasize the differences among the two models. Moreover, we added some sentences about differences in the discussion.